# Processing and Physicochemical Properties of Magnetite Nanoparticles Coated with *Curcuma longa* L. Extract

**DOI:** 10.3390/ma16083020

**Published:** 2023-04-11

**Authors:** Margarita L. Alvarado-Noguez, Ana E. Matías-Reyes, Mario Pérez-González, Sergio A. Tomás, Claudia Hernández-Aguilar, Flavio A. Domínguez-Pacheco, Jesús A. Arenas-Alatorre, Alfredo Cruz-Orea, Mauricio D. Carbajal-Tinoco, Jairo Galot-Linaldi, Elizabet Estrada-Muñiz, Libia Vega-Loyo, Jaime Santoyo-Salazar

**Affiliations:** 1Departamento de Física, Centro de Investigación y de Estudios Avanzados del Instituto Politécnico Nacional, A.P. 14-740, Ciudad de México 07360, Mexico; 2Área Académica de Matemáticas y Física, Instituto de Ciencias Básicas e Ingeniería, Universidad Autónoma del Estado de Hidalgo, Carretera Pachuca-Tulancingo Km. 4.5, Col. Carboneras, Mineral de la Reforma C.P. 42184, Hidalgo, Mexico; 3Programa en Ingeniería de Sistemas-SBAAM, SEPI-ESIME Zacatenco, Instituto Politécnico Nacional, Col. Lindavista, Ciudad de México 07738, Mexico; 4Departamento de Materia Condensada, Instituto de Física, Universidad Nacional Autónoma de México, Ciudad Universitaria, Coyoacán, Ciudad de México 04510, Mexico; 5Departamento de Toxicología, Centro de Investigación y de Estudios Avanzados del Instituto Politécnico Nacional, A.P. 14-740, Ciudad de México 07360, Mexico

**Keywords:** magnetite, nanoparticles, *Curcuma longa* L., single magnetic domains, SPIONs, XPS

## Abstract

In this work, *Curcuma longa* L. extract has been used in the synthesis and direct coating of magnetite (Fe_3_O_4_) nanoparticles ~12 nm, providing a surface layer of polyphenol groups (–OH and –COOH). This contributes to the development of nanocarriers and triggers different bio-applications. *Curcuma longa* L. is part of the ginger family (Zingiberaceae); the extracts of this plant contain a polyphenol structure compound, and it has an affinity to be linked to Fe ions. The nanoparticles’ magnetization obtained corresponded to close hysteresis loop M_s_ = 8.81 emu/g, coercive field H_c_ = 26.67 Oe, and low remanence energy as iron oxide superparamagnetic nanoparticles (SPIONs). Furthermore, the synthesized nanoparticles (G-M@T) showed tunable single magnetic domain interactions with uniaxial anisotropy as addressable cores at 90–180°. Surface analysis revealed characteristic peaks of Fe 2p, O 1s, and C 1s. From the last one, it was possible to obtain the C–O, C=O, –OH bonds, achieving an acceptable connection with the HepG2 cell line. The G-M@T nanoparticles do not induce cell toxicity in human peripheral blood mononuclear cells or HepG2 cells in vitro, but they can increase the mitochondrial and lysosomal activity in HepG2 cells, probably related to an apoptotic cell death induction or to a stress response due to the high concentration of iron within the cell.

## 1. Introduction

Currently, iron oxide superparamagnetic nanoparticles (SPIONs) have been studied for their scientific and technological applications, such as in electronic devices, the food industry, the environment, and medical fields [1,2,3,4,5,6,7,8]. Particularly, SPIONs have proven to be core for potential medical applications due to their tunable single magnetic domains (SMDs) orientation with frequency, biocompatibility, and addressable properties [9]. SPION interaction analyses have focused on their use as a drug carrier or deliverer, nano-heaters, precession, and gyromagnetic cores, in addition to multifunctional surface shells. Novelty formulations have been developed from capped magnetite nanoparticles (MNPs) with organic layers, such as liposomes, polyethylene glycol, chitosan, polysaccharides, and polyphenols, among others, in theranostics [10,11,12,13,14]. The theranostics aim to reach the combination of two or more functions as contrast agents, drug delivery, and hyperthermia [15]. Particularly, polyphenols supply hydroxyl (–OH) and carboxyl (–COOH) links through their organic rings over SPIONs’ surface. These organic molecules increase easier triggering of physicochemical stimulation to generate a biochemical effect [16]. Functional MNPs with polyphenols standout for their Fe–OH coordination interactions with the promising addition of natural anti-tumoral, anti-carcinogenic, and antioxidant agents, featured by their low toxicity, high biodegradability, and body clearance via iron metabolism pathways [17,18].

It is possible to achieve MNP synthesis through organic bases and green synthesis, based on the use of organic materials such as biomass, plant extracts, or reducing biomolecules. The process results in organically coated surfaces with bioactive properties that are useful for biomedical applications and reducing toxicity which is an improvement over conventional iodine-based contrast agents used in magnetic resonance imaging (MRI) [19]. Particularly, direct reaction with polyphenols acts to reduce and cap agents during MNP growth. Moreover, natural polyphenols can be isolated from different kinds of spices, fruits, vegetables, beverages, and edible plants [19,20]. It has been described that natural polyphenols can modulate the activation of Nrf2 and NF−κB transcriptional pathways in the cells and can significantly affect the signaling of the mitogen-activated protein kinase (MAPK) and PI3K pathways, which are related to cancer cells proliferation and the induction of apoptosis in different malignancy cell types [21]. In addition, the assembly of polyphenols into metallic particles, such as Fe III and Cu II, could induce cell apoptosis by raising the level of toxic reactive oxygen species (ROS) via the Fenton reaction, causing DNA damage which would allow efficient delivery and controlled release in the body, enhancing their bioavailability and their therapeutic activities [22,23].

It has been reported that polyphenols interfere with the gene expression and signaling pathways associated with cancer stem cell survival, inhibiting epithelial-mesenchymal transition (EMT) and oxidative stress. Furthermore, polyphenols increase drug uptake by tumor cells, decrease drug metabolism by enzymes, reduce drug efflux, and increase the sensitivity of cancer cells to chemotherapeutic agents. Polyphenols can be categorized into several subclasses, such as catechins, flavonoids, anthocyanins, catechins, isoflavones, chalcones, curcuminoids, and phenolic acids [23].

*Curcuma longa* L., better known as turmeric, is a member of the ginger family. Turmeric has worldwide importance as a potential source of medicinal and nutritional compounds. It is used to reduce inflammation, pimples and wounds, gastrointestinal problems, to lower blood glucose levels, and has antifungal properties. Particularly, the rhizome decoction is administered as an anthelmintic against jaundice as a blood purifier and to control liver disorders [24,25,26,27]. Turmeric extract (TE) can reduce the generation of intracellular ROS, inhibit cell migration, induce cell and nuclear morphological changes, and arrest the cell cycle at the S and G2/M phases resulting in cell apoptosis [27].

TE from *Curcuma longa* L. contains three diarylheptanoids called curcumin (as major component ~77%), demethoxycurcumin (~17%), and bisdemethoxycurcumin (~3%). The international standard name of curcumin (C_21_H_20_O_6_) is 1,7-bis–(4-hydroxy-3-methoxyphenyl)-1, 6-heptadiene-3, 5-dione [17,28].

Curcumin has been widely reported as a potent anti-tumoral agent, an antidepressant, and to produce biological effects such as analgesic, antiallergic, and hepato-protector, among others [29,30,31,32,33]. Curcumin is a natural polyphenolic phytochemical that selectively induces apoptosis in highly proliferating cells, as is the case of cancer cells [34,35]. It has been reported that curcumin has inhibitory effects on AXL (cell surface receptor tyrosine kinase) expressions in non-small lung cancer cells, and it modulates many signal transduction pathways involved in survival, carcinogenesis, and apoptosis [36]. In addition to curcumin, demethoxycurcumin and bisdemethoxycurcumin have been recognized as anti-tumor, anti-cancer, anti-inflammatory, and effective antioxidants, reporting antioxidant activities of 81.98%, 81.77%, and 73%, respectively [37,38,39,40,41].

The U.S. Food and Drug Administration (FDA) has classified curcumin among substances that are generally recognized as safe (GRAS) [42]. However, low water solubility and poor ingested absorption limit their direct application [43]. In recent years, advances have been made in formulating curcumin with nanocarriers to overcome its limitations by being encapsulated or by binding curcumin with other nontoxic, biocompatible materials. As is the case of Nigam et al., who reported in 2013 liposomes with curcumin-coated magnetic nanoparticles as a drug-delivery system, demonstrate their in vitro anti-cancer effects in the cervical cancer cell line (HeLa) [34]. These processes improve the hydrosolubility, absorption, stability, bioavailability, and overall therapeutic potential of curcumin [17].

The phytochemicals add organic chains, which play a major role in reducing precursor salts to nanomaterials [44]. The mechanism to induce the direct formation of Fe_3_O_4_ covered is described as follows: FeCl_2_•4H_2_O and FeCl_3_•6H_2_O were used as precursors in the synthesis, then the base KOH was added, which acts as a reducer and co-precipitation agent. At basic pH (>9), –OH groups oxidize, donating their electrons and forming a complex with metallic cations, reducing Fe^2+^/Fe^3+^ ions [19,45]. Finally, TE is added to the reaction reaching a saturated pH (10–14) and a black-color solution.

It has been reported that, from pH = 10 up to 14, Fe_3_O_4_ nanoparticles can be synthesized by a co-precipitation route with Fe^3+^ to Fe^2+^ ratio of 2:1 in the presence of non-oxidizing conditions [1,46,47,48]. Furthermore, a systematic study reports that when the pH increases, the size decreases [1].

The delocalized electrons present in the aromatic ring of TE are donated too, potentiating the reduction reaction, and due to the hydroxyl –OH or carboxyl –COOH bonds present in their polyphenols, the direct organic surface encapsulation of magnetite is favored by a covalent iron binding via ortho-dihydroxy (catechol) or trihydroxy benzene [19,49].
(1)$${Fe}^{2+}+{2Fe}^{3+}+{8OH}^{-}\to {Fe}_{3}O_{4}+{4H}_{2}O$$
(2)$$(KOH+TE)+H_{2}O_{\left(L\right)}+{2Fe}^{3+}+{Fe}^{2+}stirring\to \left[\frac{KOH+TE}{{2Fe}^{3+}:{Fe}^{2+}}\right]+{4H}_{2}O$$

During the reduction, the organic acids carboxyl groups (–COOH) lose their hydrogen atom and become carboxylate ions (COO–). This mechanism stabilizes the electrosteric attachment to the surface of the nanoparticle through organic chains [3,19].

Due to this coating, the secondary effects of the Fe_3_O_4_ nanostructures are reduced, being the organic chain congestion is the key to controlling the final size and morphology of MNPs by controlling alkaline pH (10–14), temperature 70 °C, inert atmosphere N_2_, and polyphenol volume concentration [1,50,51].

In this work, the co-precipitation route from the iron salts was improved to obtain MNPs coated directly with TE (Figure 1a), the synthesis of samples was verified, and finally, their properties were evaluated as well as their cell viability in HepG2 and PBMCs lines.

## 2. Materials and Methods

### 2.1. Materials

Potassium hydroxide (KOH) was obtained from Merck (Darmstadt, Germany). Hydrochloric acid (HCl) was purchased from Sigma-Aldrich (Rio de Janeiro, Brazil). Ferric dichloride tetrahydrate (FeCl_2_•4H_2_O) (reagent grade, p.a., ≥99%) was obtained from J.T. Baker, Inc. (Phillipsburg, NJ, USA). Ferric trichloride hexahydrate (FeCl_3_•6H_2_O) (reagent grade, p.a., ≥97%) and Tetraethylammonium hydroxide ((C_2_H_5_)_4_N(OH)) as an aqueous solution 20% wt) were purchased from Sigma Chemical Co. (St. Louis, MO, USA). All reagents were used as given without further purification. These materials were used for the reference synthesis [45]. Commercial turmeric powder from India (TRS, *Curcuma longa* L. Powder) was used in this study. Deionized water and ethanol were used in both experiments.

Chemicals such as dimethyl sulfoxide (DMSO) (Cat. D2650; 100% purity), Neutral red (NR), and 3-[4,5-dimethylthiazol-2-yl]-2,5 diphenyl tetrazolium bromide tetrazolium (MTT) were obtained from Sigma Chemical Co. (St. Louis, MO, USA); RPMI-1640 and DMEM medium, Fetal bovine serum (FBS), phytohemagglutinin, nonessential amino acids (100 mM), L-glutamine, sodium pyruvate, and antimycotic solutions were obtained from Invitrogen-Gibco (Carlsbad, CA, USA). Other reagents were obtained from J.T. Baker, Inc. (Deventer, the Netherlands), as indicated.

### 2.2. Methods

#### 2.2.1. Turmeric Extract Preparation

For this extraction, the method described by Alvis et al., 2012 was used with the purpose of preserving the highest concentration of phenolic (curcumin, desmethoxycurcumin, and bisdemethoxycurcumin) and antioxidant compounds, maintaining its free radical scavenger activity in the resulting extract [52].

First, 20 g of commercial turmeric powder was mixed in 400 mL of ethanol 79%, heated to 200 °C, stirred magnetically at 400 rpm for 20 min, and allowed to cool [52].

#### 2.2.2. Co-Precipitation Route from Iron Salts for Bare MNPs

The chemical synthesis of MNPs was performed according to the co-precipitation method of iron salts [45] with some modifications. The synthesis procedure was as follows:

First, 6.96 g of FeCl_3_•6H_2_O and 2.52 g of FeCl_2_•4H_2_O were separately mixed with 25 and 6.25 mL of degassed water, respectively. A 3-necked flask filled with nitrogen (N_2_) was heated in a water bath until it reached a stable temperature of 70 °C. Then, 1.25 mL of the FeCl_2_•4H_2_O solution was poured into the 3-necked flask, followed by the addition of 5 mL of the FeCl_3_•6H_2_O solution, and stirred mechanically at 200 rpm for 15 min.

Then, 50 mL of (C_2_H_5_)_4_N(OH) was placed in a beaker with an N_2_ atmosphere for 5 min. The base was added by drops to the 3-necked flask, mechanically stirred at 200 rpm, and heated at 70 °C for 10 min while maintaining the inert atmosphere to avoid oxidation.

The resulting solution (black color, pH = 14) was poured into a beaker allowing it to cool and removing the supernatant. Then, 50 mL of ethanol and 50 mL of deionized water were added, sonicating the mixture for 5 min and letting the solution container rest on a magnet until the sample precipitated.

Once the sample precipitated, a second and third wash using an ethanol/water solution under the same experimental conditions was performed. Furthermore, four more washes were carried out with double distilled water, and then the sample was lyophilized to achieve maximum moisture extraction.

The MNP powder obtained was sensitive to the magnetic field. The powder was subsequently stored under vacuum conditions.

#### 2.2.3. Synthesis of MNPs Coated Turmeric Extract Base, G-M@T

As in the co-precipitation route from iron salts for bare MNPs, the synthesis initiates using 6.96 g of FeCl_3_•6H_2_O and 2.52 g of FeCl_2_•4H_2_O mixed separately during 30 min with distilled water in a molarity of 0.02 M and 0.01 M, respectively. N_2_ was injected in a 3-necked flask during all reaction procedures and heated at 70° C in a water bath. Then, a ½ Fe^+2^:Fe^+3^ solution was poured into the 3-necked flask and stirred mechanically at 200 rpm for 15 min. An amount of 10 mL of 7 M KOH in deionized water was added to the 3-necked flask by dropping.

After 10 min, 50 mL of TE was dropped in the solution, reaching pH = 14 and a black-color solution. The synthesis was poured into a beaker allowing it to cool by removing the supernatant. Next, 50 mL of ethanol and 50 mL of deionized water were added, sonicating the mixture for 5 min and letting the solution container rest on a magnet until the sample precipitated. Furthermore, two washes using ethanol/water solution were performed, and four more washes were carried out with double distilled water, lyophilizing the sample at the end and being stored under vacuum conditions. The G-M@T powder obtained was also sensitive to the magnetic field.

### 2.3. Physicochemical Characterization

#### 2.3.1. X-ray Diffraction

X-ray diffraction (XRD) patterns of G-M@T powder were obtained in an X-ray diffractometer (Siemens model D5000, Munchen, Germany) equipped with a CuK_α_ radiation wavelength of 1.5406 Å operating at 35 kV acceleration voltage and current of 25 mA. The average crystallite size was estimated using PowderCell 2.3 software (JCPDS PDF-19-0629) [53] with a lattice parameter of 8.396 Å, inverse space group *Fd*$\overset{-}{3}$*m* (227), and inverse spinel FCC structure.

#### 2.3.2. Fourier-Transform Infrared Spectroscopy

The samples were measured using Fourier-transform infrared spectroscopy (FT-IR) in the range of 4000–450 cm^−1^. The superficial characterization resulted in the reflection spectrum of the bands of the functional groups of the inorganic and organic substances. To identify the elements contained in the analyzed samples, a NICOLET brand FT-IR equipment, Model 6700, was used in the transmission mode.

#### 2.3.3. X-ray Photoelectron Spectroscopy

An X-ray photoelectron spectroscopy analysis (XPS) was performed using a Thermo Scientific (East Grinstead, UK) K-alpha spectrometer. A monochromatized X-ray beam from an Al anode was focused on the sample surface having a spot size of 400 µm and an energy of 1486.6 eV. The powder samples were placed on double-sided carbon tape and crushed to reveal clean surfaces. In order to minimize the interaction with air, the materials were immediately transferred to the load-lock and degassed to pressures close to 1 × 10^−8^ Torr for 48 h; subsequently, they were introduced into the analysis chamber where measurements were taken at a residual pressure of approximately 1 × 10^−9^ Torr. During the analysis, a charge neutralizer was used to control charging effects. All the spectra were calibrated using the adventitious C 1s peak at a binding energy (BE) of 284.6 eV using the 5.9 Avantage software version.

#### 2.3.4. Photoacoustic Spectroscopy

Using the photoacoustic spectroscopy (PAS) technique, it was possible to obtain the sample optical absorption spectrum. A homemade photoacoustic spectrometer was used, which has a Xenon lamp as a source of radiation, followed by a monochromator and a mechanical chopper fixed at 17 Hz. The monochromatic and modulated light is then guided by an optical fiber to the photoacoustic cell, which contains the sample to be analyzed, and this cell is sealed hermetically by vacuum grease. The air pressure variations are associated with the periodic heating, related to the optical absorption from incident light in the sample. The response is detected by a microphone located inside the cell, and its electrical signal is amplified by a lock-in amplifier. Finally, the photoacoustic signal, as a function of the incident light wavelength, is stored in a computer for its analysis. In addition, the band gap energy (Eg) of the synthesized samples was estimated using Tauc’s equation:(3)$${\left(\alpha hv\right)}^{n}=B\left(hv-E_{g}\right)$$
where α is the absorption coefficient (which is proportional to the optical absorption spectrum measured by PAS), *hν* is the photon energy, *n* is the type of electronic transition (for magnetite, a direct electronic transition is expected, then *n* = 2 was used), and *B* is a constant [54].

#### 2.3.5. Transmission Electron Microscopy

The morphology and size of the nanoparticles (NPs) were obtained by transmission electron microscopy (TEM) using a JEOL (Tokyo, Japan) JEM2010 microscope equipped with a LaB_6_ filament at 200 keV. The suspension of nanoparticles in distilled water was sonicated for 10 min and then dropped on a copper grid lacey 400 mesh. The average nanoparticle size was determined by measuring 150 particles. Their distribution was fitted to a Gaussian function.

#### 2.3.6. Superconducting Quantum Interference Device Magnetometer

The magnetic response of the nanoparticles was carried out using a Superconducting Quantum Interference Device (SQUID) Magnetometer model MPMS-3 DC & VSM brand Quantum Design under ambient temperature conditions.

#### 2.3.7. Topography and Magnetic Domains

The magnetic domains and topography analysis of the surface of the nanoparticles were performed using Scanning Probe Microscope JEOL, JSPM-5200 in Magnetic Force Microscopy mode (MFM). All powder samples were confined separately in carbon adhesive tape. A magnetic tip NSC18, Co-Cr/Al Micromasch, with an uncoated radius of 8 nm, coated radius <60 nm, and full tip cone angle of 40°, was used. This tip was magnetized with a neodymium magnet 5 min prior to the characterization. The 2D and 3D images, profiles, and domain measurements were processed with Gwyddion 2.62 software.

### 2.4. Biological Evaluation

#### 2.4.1. Peripheral Blood Mononuclear Cells and HepG2 Cell Line

Heparinized human blood was collected by venipuncture from male volunteers (24–28 years old) with no history of degenerative disease and who did not smoke, drink, or take any drug for at least one month before the donation. The study was conducted in accordance with the Declaration of Helsinki, and the procedure was approved by the institutional bioethics committee for studies on humans of CINVESTAV (COBISH) (protocol 045/2017). We placed blood samples in Falcon tubes to isolate PBMC by centrifugation at 2000 rpm for 25 min on Hypaque/Ficoll 400 solution (Histopaque^®^-1077 solution, Cat. num. 10771; 1.077 g/mL density; Sigma Aldrich, St. Louis, MO, USA), as described elsewhere [55].

After isolation, cells were suspended in fresh RPMI medium supplemented with 10% FBS, 1% nonessential amino acids (100 mM), and 1% L-glutamine (2 mM) and cultured on 96 well plates at 2 × 10^5^ cells/mL. A second set of cells was cultured with phytohemagglutinin stimulation (PHA; 5 μg/mL) for 24 h before treatments were applied. The cells were incubated in a humidified chamber at 37 °C with a 5% CO_2_ atmosphere for 24 h.

The human cell line was HepG2 (hepatocellular carcinoma, ATCC) [56]. These cells were cultured in DMEM medium supplemented with 10% FBS, 1% nonessential amino acids (100 mM), 1% sodium pyruvate (100 mM), and 1% antibiotic-antimycotic solution (100×) in a humidified chamber at 37 °C and 5% CO_2_. Trypsin (0.25 mg/mL) was added, incubated for 2 min, and then inactivated by adding supplemented medium; the cell suspensions were centrifugated at 1500 rpm for 5 min, all to subculture cells. Cells were resuspended in a fresh medium, and then the cell density was adjusted and cultured on 96 well plates at 2.5 × 10^4^ cells/mL and incubated as previously mentioned for 24 h.

#### 2.4.2. Treatments and Cell Viability using MTT and Neutral Red Uptake (NR) Assays

Stock solutions and serial dilutions of MNPs (uncoated) and Fe_3_O_4_ coated with TE labeled as G-M@T at 0. 12.5, 25, 50, and 100 μg/mL were prepared in DMEM or RPMI medium using an ultrasonic Processor (GEX 130 PB) for 30 sec (130 watts, 20 KHz with 40% ampl). A stock solution of TE was prepared in DMSO, and serial dilutions were made in DMEM or RPMI at 0, 3.125, 6.25, 12.50, 25, 50, and 100 g/mL.

Serial dilutions of MNPs and G-M@T (previously sonicated) and TE were immediately added to cultures and incubated for 48 h. Cell viability was determined using the reduction of MTT to formazan and neutral red (NR) uptake assays. MTT solution (5 mg/mL in PBS (137 mM NaCl, 2.7 mM KCl, 10.1 mM Na_2_HPO_4_, 1.8 mM KH_2_PO_4_, pH 7.4)) was added to 96-well plates and incubated for 3 h (PBMCs) or 2 h (HepG2). After that, the medium was discarded, and the formazan crystal were dissolved with DMSO (100 μL); then, the absorbance of the sample was determined on an ELISA-plate reader (Multiskan FC, Thermo Scientific) at 490 and 520 nm [57].

In the NR uptake assay, NR solution (0.34 mg/mL in PBS, pH 6) was added to the cultures, and the plates were incubated for 3 h (PBMCs or HepG2). After that, the medium was discarded, the cells were washed twice with PBS, 100 μL of an acidic solution (water/ethanol/acetic acid 49:50:1 v/v/v) was added to dissolve the NR inside the cells, and the absorbance at 540 nm was determined. The values were adjusted with the values obtained with the vehicle treatment (DMSO 0.0125%) that were considered 100% viable.

#### 2.4.3. Statistical Analysis

Data are shown as means ± standard deviation (SD) of at least three independent experiments performed in triplicate. We chose *p* < 0.05 as statistically significant, using the one-way ANOVA post hoc Bonferroni or Student’s *t*-test, as appropriate.

## 3. Results

Following the procedures described in Section 2, two types of nanoparticles were obtained. MNPs correspond to magnetite nanoparticles obtained by the co-precipitation iron salts method (used as reference) and G-M@T, which corresponds to magnetite organic base synthesis covered with TE (G-M@T).

### 3.1. XRD Analysis

The diffractograms obtained for MNPs and G-M@T are presented in Figure 1b. The nanoparticles contributions showed the main magnetite peaks at 2θ positions of 30° (220), 35.19° (311), 43° (400), 53° (422), 57° (511), and 62° (440), matching with the standard JCPDS card no. 19-0629 related to FCC cubic spinel inverse crystalline structure [58]. In the G-M@T sample, it is possible to observe the MNPs plus the TE amorphous coating contributions. The peaks of the G-M@T sample were found to be slightly displaced towards maghemite (γ−Fe_2_O_3_) originated surface oxidation during the synthesis procedure [19]. This generated a lattice parameter distortion, *a* = 8.344 Å [59]. Debye-Scherrer equation was used to define the mean particle sizes <*d*> for the MNPs and G-M@T samples (Table 1).

### 3.2. FT-IR Analysis

The FT-IR analysis was performed to characterize the presence of iron oxide bonds, the structural composition of the coating, and its bonding on the surface, showing in the spectra (Figure 1c) the transmittance of MNPs, G-M@T, and Turmeric powder.

In MNPs and G-M@T samples, it was confirmed that the iron presence through the observed peak in 572 cm^−1^ was attributed to Fe–O bond vibration [4,60,61].

The organic responses in Turmeric samples were defined in the region between 1150 and 930 cm^−1^, coinciding with the bending vibrations of C–H in the plane of the pyranose ring and the stretching vibrations of C–O [34]. Bands around 1150, 1105, and 1080 cm^−1^ are part of the starch response and are associated with the glucose C–O bond vibrations of the primary and secondary alcohol groups [62]. It is possible to observe a band in 1640 cm^−1^ associated with carbonyl groups due to the organic base used in the chemical synthesis (uncoated MNPs), while peaks in the range of 1430–1650 cm^−1^ are attributed to the ring (C–N, C=N) stretching vibrations [19,63]. The peaks located between 2925 and 2850 cm^−1^ belong to the deformation of C–H [60]. The organic component and the presence of hydroxyl groups on the surface of the particles can be confirmed by the band around 3400 cm^−1^, attributed to the deformation of –OH [19,60].

In G-M@T and turmeric samples, a peak was observed placed at 1739 cm^−1,^ which is related to the vibrations of the C=O bond of the fatty acid chains, and which could occur by curcumin degradation [34,64]. It has been reported that curcumin exhibits characteristic peaks located at 3288 cm^−1^, 2930 cm^−1^, 1634 cm^−1^, 1515 cm^−1^, and 993 cm^−1^, which are related to their phenolic O–H stretching vibration, C–H stretching, aromatic moiety C=C stretching, and for the last two peaks C–O–C stretching [64]. Moreover, the polyphenolic surface contribution was related to curcumin in the absorption at 1581, 1511, 1279, and 1152 cm^−1^. These wave vibrations were defined for the benzene ring, C=C vibrations, aromatic C–O stretching, and C–O–C stretching modes, respectively [62,65].

**Figure 1 materials-16-03020-f001:**
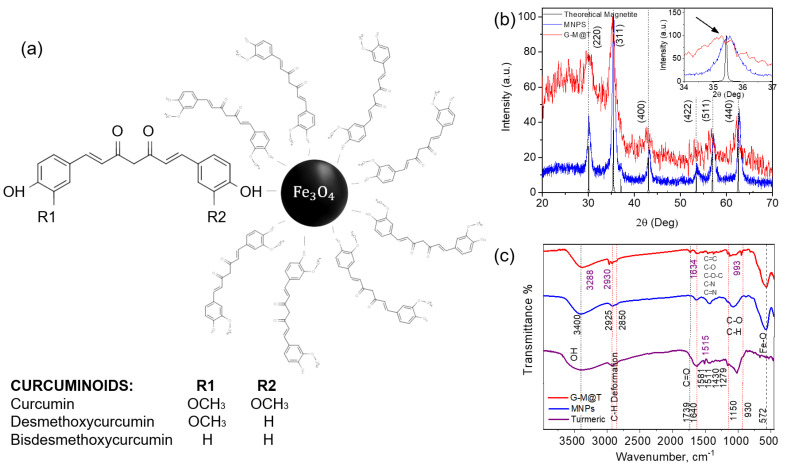
Configuration of Fe_3_O_4_ synthesized covered by *Curcuma longa* L. extract C_21_H_20_O_6_; (**a**) Curcumin bond –OH over MNPs. The polyphenolic coating was formed directly during the synthesis processing. Nature of bonding was covalent with Fe ions [66,67]. (**b**) XRD shows MNPs (blue line), G-M@T (red line), and the theoretical diffraction pattern of magnetite (black positions). The inset reveals a slight displacement of the peak corresponding to the (311) plane. The curve under the red line indicates the addition of C_21_H_20_O_6_ over the MNPs surface. This had an amorphous response. (**c**) FT-IR spectra of the synthesized samples. It shows the presence of Fe–O and aromatic compounds on the nanoparticles’ surface.

### 3.3. XPS Analysis

The surface chemistry of the turmeric powder, MNP, and G-M@T samples were systematically analyzed by XPS. Several photoelectron signals related to the Fe 2p, C 1s, and O 1s states were found.

The high-resolution XPS spectra of the C 1s and O 1s states for the three samples are displayed in Figure 2.

The C 1s spectrum of turmeric powder was deconvolved into four components. Since it has been claimed that for this kind of compound, the C 1s components have a rich structure, the peak placed at 284.6 eV has been assigned to C=C, the peak at 286.1 eV to C=O, C−C and C−H bonds, and the photoelectron line at 287.9 eV could be ascribed to C–N or C–OH bonds (see Figure 2a) [68,69]. An additional peak has been reported for curcumin at 292.9 eV, which is attributed to a π − π * shake-up due to excitations between filled and empty conjugated π states of the aromatic ring [69]. The core-level spectrum of the C 1s state for MNPs is shown in Figure 2b). The peak at 284.6 eV has been assigned to C–H bonds, while the peak at 286.1 eV has been attributed to C=O bonds. The signal at approximately 288 eV could be related to O–C=O [70,71]. In the G-M@T sample, a superposition of peaks from turmeric powder and Fe_3_O_4_ was observed; in particular, the π − π * shake-up from turmeric powder is clearly visible (see Figure 2c).

The O 1s spectrum of turmeric powder was fitted using two peaks: the first one at 531.0 eV could be related to oxygen atoms or O=C bonds [68,69], while the other one at 532.6 eV has been assigned to O–C or O–H bonding, Figure 2d [68]. The O 1s core-level spectrum of MNPs is shown in Figure 2e. The main peak below 530 eV is associated with lattice oxygen (0^2−^) [71]. Fujii et al. (1999) showed only one peak for the O 1s signal as a sign of a contaminant-free surface [72]. In our case, more features appeared at higher binding energies. These peaks have been related to OH^−^ (~531 eV) and C–O or O–C=O species (~532 eV) adsorbed on the surface [71]. The origin of chemisorbed hydroxyl or carbonyl species is due to both the synthesis method and subsequent manipulation under atmospheric conditions. In the G-M@T sample, the peaks displayed a behavior similar to that presented by MNPs; however, some signals from turmeric powder could be overlapped with the peak at ~531 eV (see Figure 2f).

The survey spectra for turmeric powder, MNPs, and G-M@T samples are displayed in Figure 3a. For turmeric powder, the main peaks placed at approximately 284.6 and 532.5 eV are attributed to carbon and oxygen (C 1s and O 1s), respectively. On the other side, MNPs and G-M@T display C 1s, O 1s, and Fe 2p signals.

The Fe 2p orbitals for MNP and G-M@T samples are displayed in Figure 3b,c, respectively. In order to deconvolve these signals, different constraints based on quantum mechanical properties were considered, including the peak-to-peak area ratio, spin-orbit splitting, and Full-Width at Half-Maximum (FWHM) [73]. In particular, the peak-to-peak area ratio for 2p states satisfies the condition Area(Fe 2p_1/2_)/Area(Fe 2p_3/2_) = 1/2 [74]. Each spectrum was fitted considering two oxidation states for Fe compounds, Fe 2p^2+^ and Fe 2p^3+^ since iron cations in maghemite (γ−Fe_2_O_3_) and hematite (α−Fe_2_O_3_) have been reported in the 3+ state while both states are present in magnetite (Fe_3_O_4_) [75]. In addition, two shake-up satellite doublets for the 2+ and 3+ oxidation states were considered in the analysis. As observed in Figure 3b,c, both spectra presented similar features. The Fe 2p^2+^_3/2_ and Fe 2p^2+^_1/2_ components for the MNPs were found at 709.9 and 722.7 eV, respectively. Their corresponding satellites appeared at 714.4 and 728 eV. On the other hand, the Fe 2p^3+^_3/2_ and Fe 2p^3+^_1/2_ signals were observed at 710.8 and 724.3 eV, followed by the shake-up satellites at 719.1 and 732.8 eV. For the G-M@T combined system (turmeric powder with iron oxides), the Fe 2p^2+^_3/2_ and Fe 2p^2+^_1/2_ peaks were displayed at 710.0 and 722.9 eV, while their 2+ satellites appeared at 714.4 and 728.0 eV. For the 3+ states, the Fe 2p^3+^_3/2_ and Fe 2p^3+^_1/2_ signals were found at 710.9 and 724.4 eV, with their satellites being placed at 719.2 and 732.8 eV. For both MNPs and G-M@T samples, the spin-orbit separation was close to 12.8–12.9 eV for the 2p^2+^ contributions and 13.5 eV for the 2p^3+^ states. It should be noticed that the 3+ components are more intense than the 2+ components, indicating that the main iron oxide phase is magnetite (Fe_3_O_4_) [76].

The XPS valence band spectra for MNPs and G-M@T samples are presented in Figure 3d. In this case, the low energy band placed between 0–10 eV has been attributed to bulk crystals, while the peak at approximately 21.7 eV represents the O 2s level [72]. It has been reported that for α and γ−Fe_2_O_3_ phases, a satellite band is distinguishable between 10–20 eV, while Fe_3_O_4_ does not exhibit this band. In our case, from the inspection of the MNPs and G-M@T signals, it could be suggested that the main iron oxide phase is magnetite, with a slight oxidation (mostly seen in G-M@T nanoparticles), as previously observed from the core-level analysis of the Fe 2p signals.

### 3.4. PAS Analysis

The bandgap energies of G-M@T and MNPs were computed by using Equation (4), yielding 1.26 and 1.24 eV, respectively. These values are in agreement with those reported by El Ghandoor et al., 2012 [77].

### 3.5. TE Coating, Particle Size, and Electron Diffraction

The semi-spherical morphology and nanometric size of the magnetite nanoparticles are observed from TEM images in Figure 4a,b,e,f. Selected area electron diffraction (SAED) analysis showed rings characteristic of the high crystalline structure of the samples (see Figure 4c,g with the contribution of TE coating plus Fe_3_O_4_ NPs cores). In addition, the XRD diffractograms of MNPs and G-M@T matched very well with their respective SAED patterns.

The organic shell thickness was measured over the low contrast areas due to the amorphous surface layer (~3 nm) corresponding to the TE polyphenols, as shown in Figure 4f. Histograms show the particle size distribution, considering the analysis of 150 particles in both cases. The mean particle size <Ø> was calculated from the Gaussian fitting, resulting in Ø = 12.58 ± 1.17 nm for MNPs and Ø = 10.65 ± 1.49 nm for M-G@T, as exhibited in Figure 4d and Figure 4h, respectively.

### 3.6. SQUID Magnetometer

The magnetic saturation (M_s_) and the applied field (H) showed a difference in the magnetic perform of MNPs due to the TE coating barrier, as plotted in Figure 5. Closed hysteresis loop cycles were obtained for MNPs and G-M@T samples. MNPs had M_s_ = 76.00 emu/g and H_c_ = 19.26 Oe according to the uneven shape and sizes 10–16 nm [78], while that for G-M@T showed M_s_ = 8.81 emu/g and H_c_ = 26.67 Oe values. In this case, the hysteresis loop was lower than bare MNPs. The semi-spherical nanoparticles were in a range of 7–16 nm with an additional TE layer as an antiferromagnetic barrier. As reported, the amorphous polyphenolic (C–H, C=C, C–O, C–O–C) cover around 3 nm formed an antiferromagnetic barrier over the surface nanoparticles [79]. Moreover, the particle size in G-M@T (10.65 nm) and lattice parameter (8.344 Å) influence the magnetic behavior [16]. TE layer has introduced a disorder over the nanoparticle surface [80]. Close hysteresis loops were near to SPIONs’ behavior. The structure, shape, particle size, and organic coating have a direct effect on the magnetic response of coated MNPs.

### 3.7. Topography and Magnetic Domains

Topography G-M@T of magnetic nanoparticles was defined in a scanned area of 300 × 300 nm by MFM in initial conditions without applied field (H0). The 3D figure shows the surface of the nanoparticle distribution and the profile indicates the sizes obtained from 10–13 nm, Figure 6.

G-M@T MFM mapping showed the magnetic flux related to magnetic field lines from SMDs at Lift High in a range of 0–8.5 nm z-axis and applied field from 0–5000 Oe. The MFM mapping was built from Zero (H0) to saturation(H↑) conditions and demagnetization from H↑ to H0. The 3D Topography and MFM responses showed well-defined nanoparticles at the surface and the random orientation of SMDs, respectively, at initial conditions Lift High, z = 0 and H0. Topography showed a contraction of G-M@T nanoparticles at H↑ and the characteristic attraction response from SMDs with a uniaxial orientation at Lift High, z = 8.5 nm and (H↑). Demagnetization showed the recovery at initial conditions z = 0 and H0. The topography indicated the nanoparticle distribution, while MFM showed the signal of SMDs with low remanence and random distribution, as in Figure 7 from left to right.

The MFM zoom at H↑ showed the magnetic field lines from SMDs due to the magnetic tip interaction with the surface [81]. The SMDs were oriented in the direction of the magnetic applied field, and parallel flux lines indicated the uniaxial anisotropy. The profile illustrates these lines estimated in ~1.2 nm over SMDs, Figure 8. This indicates the possibility of using the G-M@T MNPs as addressable nanocarriers.

### 3.8. Biological Evaluation

In addition to the physicochemical characterization, the toxicity of the uncoated MNPs and the G-M@T were analyzed. As a first step, the evaluation of the metabolic activity of the cells was quantified using the MTT assay method, which indicates the mitochondrial activity of the cells, and using the Neutral Red Uptake method, which indicates the lysosome activity. Both methods are accepted and used to determine the viability of cells in culture. The exposure to MNPs and G-M@T showed no effect on cell viability by either test on normal human PHA-stimulated PBMCs (Figure 9a,c). The exposure to TE did not modify the mitochondrial activity (Figure 9a) but slightly reduced the lysosomal activity (Figure 9c). On the other hand, exposure of HepG2 cells to TE significantly reduced cell viability as quantified by both parameters, MTT reduction (Figure 9b) and NR uptake from 50 mg/mL (Figure 9d), indicating that the curcumin-containing extract presents selective antineoplastic activity against tumoral cells.

Also, MNPs, as well as the G-M@T nanoparticles, induced a stress response in the cells as they increased the mitochondrial and lysosomal activity in the cell cultures (Figure 9b,d), suggesting that the cells could be initiating an apoptotic death response because it has been reported that the apoptotic process requires the production of ATP by the mitochondria [82] and the increased degradation of organelles by the lysosomes [83].

## 4. Discussion

MNPs were coated directly with *Curcuma longa* L. extract (G-M@T). Fe_3_O_4_ lattice distortion was observed by Fe^2+^ surface vacancies [19]. This has been associated with MNPs surface oxidation during their processing which is consistent with XPS analysis. These changes are increased as the function of particle size decreases [1]. Structural analysis was agreement by SAED and XRD. Moreover, the G-M@T XRD showed an amorphous contribution corresponding to the TE covering by scattering under Fe_3_O_4_ characteristic peaks.

Z. Hami (2020) has obtained spherical iron magnetic nanoparticles ~20 nm coated with nano curcumin and chitosan as a coating polymer with magnetic saturation of 35.3 emu/g [84]. Ramírez-Nuñez et al. (2018) reported a synthesis route to produce spherical magnetite nanoparticles 10–14 nm with *Vanilla planifolia* and *Cinnamomun verum* extracts; their magnetic saturations were 59–70 emu/g [19]. Both studies showed O–H stretching vibrations peaks in the region around 3400 and 2930 cm^−1^ by FT-IR analysis. G-M@T NPs were in agreement with the –OH bonding attached over the surface of synthesized nanoparticles in this work [19,84]. In our case, we used potassium hydroxide as alkaline media and turmeric extract in direct coating processing, obtaining the organic bands corresponding to TE extract on the NP surface in the FT-IR analysis. Additionally, it was possible to identify their energy response over their surface by XPS analysis.

As seen from the XPS analysis, iron oxides have been formed for MNPs and G-M@T signals. In particular, Fe^2+^ and Fe^3+^ oxidation states have been found in γ−Fe_2_O_3_ (maghemite), nonstoichiometric Fe_3-δ_O_4_, and Fe_3_O_4_ (magnetite) [72,75,76,85]. In the Fe_3_O_4_ phase, represented by [Fe^3+^]_tet_[Fe^2+^Fe^3+^]_oct_O_4_, the Fe^2+^ ions are placed in octahedral sites while the Fe^3+^ ions occupy both octahedral and tetrahedral structural sites [72]. The satellite peak structures, which originated from the ejection of 2p photoelectrons and attributed to the electron transition from the 3d to the empty 4s orbital, provide useful information to determine the presence of Fe^3+^ high-spin ferrous species [70,76]. In the case of Fe_3_O_4_, it is characterized by a visible tail at the low binding energy region due to Fe^2+^ multiplets [76]. Moreover, the XPS and FTIR spectroscopy analysis agreed with the MNPs cores and surface TE signals from G-M@T.

Thermogravimetric analysis (TGA) makes it possible to study the thermal stability and estimate the mass percentage of the organic shell present in the MNPs [86,87,88,89,90]. In this work, we use Transmission Electron Microscopy (TEM) to estimate the percentage of TE (shell) and Fe_3_O_4_ (core) contained in the G-M@T sample. Using the micrographs reported in this manuscript, as can be seen in Figure 4f, taking into account the reported densities of magnetite and turmeric as follows [91,92,93].

Considering the TEM analysis, the core MNPs radius (~8.56 nm) and TE thickness layer (~3 nm) in G-M@T nanoparticles (Figure 4f). The percentage in composition can be estimated indirectly from
(4)$$V_{Shell}=\frac{4}{3}\pi \left(r_{cs}^{3}-r_{c}^{3}\right)$$
(5)$$V_{Core}=\frac{4}{3}\pi \left(r_{c}^{3}\right)$$
where r_cs_ = core-shell radius (nm), r_c_ = Core radius (nm). Then, by using the density definition, with the magnetite core and the turmeric shell densities as 5.2 g/cm^3^ and 1.002 g⁄cm^3,^ respectively, the core-shell total mass can be obtained [94]:(6)$$m_{Total}=m_{shell}+m_{core}$$

Then, by using Equations (4) and (5), the core and shell masses can be obtained, and finally, the estimated composition percentages for G-M@T were: $m_{Total}=100\%,m_{core}=75.32\%,and\;m_{shell}=24.66\%$.

Concerning the magnetic properties, G-M@T NPs showed tunable SMDs interactions with uniaxial anisotropy as addressable cores at 90–180° due to their superparamagnetic behavior [81]. It is known that magnetization depends on the crystallinity of the nanoparticles, where the higher the crystallinity of magnetite NPs, the higher the magnetization [2]. From the XRD diffractogram (Figure 1b), it can be observed the crystallinity of the magnetite in the MNPs sample (Ms = 76.00 emu/g), while in the G-M@T sample, the crystallinity of the magnetite and an amorphous contribution is observable due to the organic TE cover (Ms = 8.81 emu/g). This Ms reduction in the G-M@T sample is due to the polyphenol layer around the magnetite core since it has been reported that a decrease in Ms at smaller sizes is attributed to pronounced surface effects [2,19,77]. The TE layer behaves like a diamagnetic barrier; hence, when the coating thickness increased, the saturation magnetization decreased [47].

Nanoparticles may be susceptible to aggregate due to the Van der Waals interaction [95]. This aggregation is undesirable, especially in NPs seeking its application in the biological area, because it can block capillaries or reduce the probability of reaching the therapeutic objective [96]. Electrostatic and steric repulsion can prevent nanoparticle aggregation by stabilizing the suspension [95]. In our G-M@T system, hydroxyl and carboxyl groups mainly promote suspension stability due to electrostatic repulsion [17,97,98]. The pH of the suspension changes the net electrical charge of the NPs due to deprotonation mechanisms [97,99]. The measurement of the Zeta potential is a method that allows the determination of the suspension stability by electrostatic repulsion, even depending on the pH of the suspension. Some authors suggest that a Zeta potential of >±20 mV is desirable to avoid the aggregation of NPs [100]. It has been reported that MNPs covered with curcumin have a zeta potential >±20 mV in a pH range of 2–12, where the iso-electric point is pH~4 [17,97,98,100,101,102,103].

By the phenomena described above, the stability of the suspension is expected. Additionally, stability analysis of the synthesis solution was performed by dynamic magnetic precipitation [104,105]. G-M@T and MNPs were suspended in 20 mL of deionized water, and a neodymium magnet was placed next to them. Photographs were taken as a function of time at t = 0, 1, 5, 10, 15, 20, 24, 30, and 36 h (see Figure 10). Both samples were stable and observed at room temperature. Several reports have been done to show that these kinds of magnetic nanoparticles, covered with phytocompounds, such as terpenoids, polyphenols, flavonoids, and prominent functional groups, offer robust shielding on the NPs surface to prevent their aggregation and keep them stable in suspension, enhancing their hydrophilicity and biocompatibility [44,47,48,103].

The bandgap energy of G-M@T was in the range reported in the literature, where it is described that the energy band gap of the nanomaterials is inversely proportional to the nanoparticle sizes and average crystallinity volume size [54,77]. El Ghandoor et al. (2012) found a Fe_3_O_4_ bandgap of 1.92 eV in bare MNPs of 10 nm, while the obtained band gap was around 1.2 eV in G-M@T for 12 nm, confirming this statement [54,77,106]. The energy band gap obtained also shows a behavior of a semiconductor material with an absorption edge in the infrared region (~1000 nm) [81]. Recently, photothermal therapy (PTT) and magnetite nanoparticles have been simultaneously used to improve the cancer cell-killing effect by apoptosis [7,8,107]. Usually, near-infrared (NIR) light is used in photodynamic therapy (PDT) since it provides deep tissue penetration with minimum nonselective cell death, and by using light-absorbing agents, it is possible to generate reactive oxygen species (ROS) such as H_2_O_2_ [7,8]. It has been reported by Wu et al. (2022) that by using urchin-like hollow magnetic microspheres loaded with a photosensitizer (Chlorin e6, Ce6), irradiated by NIR light, ROS were generated, improving the cancer cells killing effect [7]. On the other hand, Ramezani et al. (2022) reported the use of PDT with curcumin as a photosensitizer, which enhances ROS by stimulating apoptosis and autophagy to reduce the viability of cervical cancer cells [108]. These reports of the simultaneous use of PDT and MNPs motivate the potential use of our synthesized nanoparticles as a more effective photothermal/magnetic agent against cancer cells.

The polyphenol cover in the G-M@T nanoparticles did not produce intrinsic toxicity in normal human white blood cells (PBMCs) but suggested the increase of metabolic stress on transformed human hepatic cells (HepG2), indicating that they could be useful as carrier systems in drug delivery with antineoplastic compounds. Nosrati and co-workers (2018) developed Bovine Serum Albumin (BSA) coated iron oxide magnetic nanoparticles with curcumin, founding in the MTT assays that they are highly biocompatible and do not possess a toxic effect, however, on MCF-7 cells corresponding to human breast cancer cells, it was revealed the significant cytotoxicity of nanoparticles, agreeing with our results. Ramezani et al. (2022) synthesized Fe_3_O_4_ nanoparticles coated with hyperbranched polyglycerol (HPG) and folic acid (FA), being functionalized and loaded with curcumin. The MTT assay revealed their high cytotoxicity on HeLa cells, reducing the viability of cancer cells after 24, 48, and 72 h [108]. Stands out the magnetic nanoparticles covered with curcumin as a proper material for further in vivo applications [109].

Moreover, the Fe ions of the Fe_3_O_4_ nanoparticle surface can react with hydrogen peroxide (H_2_O_2_), producing hydroxyl (–OH), inducing a potent inhibition effect against cancer cells by a process called the Fenton process [107,110,111]. Due to the intense metabolic activities of cancer cells, their microenvironment is characterized by a high level of H_2_O_2_, and an acid pH, which allows selective cancer cells death with regard to normal cells, considering that the toxicity of the –OH radical must reach a certain level threshold to achieve sufficient therapeutic efficacy. Otherwise, the radical may become carcinogenic and cause further proliferation of cancer cells [110]. In this sense, the study of the Fenton effect in iron nanoparticles covered with turmeric extract applied to cancer cells is proposed as a perspective.

Many authors have used plant or leaf extracts to obtain magnetite nanoparticles, being the best for biological applications in the size range of 10–50 nm, reporting a spherical shape and magnetic saturation around 30–70 emu/g [19,84,112]. In our case, our shapes and sizes obtained are in the range reported, despite having less magnetic saturation, G-M@T still being biocompatible and being adequate for use in biological applications.

The TE encapsulation allows its use against metabolic disorders, cardio- and vaso-protective, hepato-protective, neuroprotective, and antidepressant, mediating anti-inflammatory and antioxidant activities [113]. Several future studies could be carried out controlling the size, shape, and thickness of the TE shell for its development in theranostics [2,4,114,115,116].

## 5. Conclusions

In this study, we developed an easy and straightforward method for synthesizing magnetite nanoparticles loaded with turmeric extract to diminish the pollutant waste products generated by conventional synthesis.

G-M@T nanoparticles of ~12 nm, superparamagnetic response with uniaxial orientation, and –OH surface bonds did not produce intrinsic toxicity in normal human white blood cells (PBMCs) but suggested the increase of metabolic stress on transformed hepatic human cells (HepG2). Several nanocarriers could be tailored from core-shell systems from polyphenols and MNPs as an alternative treatment for different illnesses.

## Figures and Tables

**Figure 2 materials-16-03020-f002:**
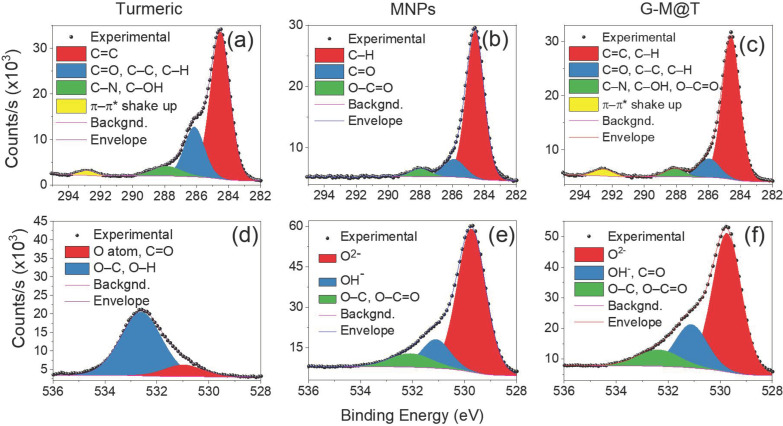
High-resolution XPS spectra of the C 1s (**a**–**c**) and O 1s (**d**–**f**) orbitals for turmeric powder, MNPs, and G-M@T samples.

**Figure 3 materials-16-03020-f003:**
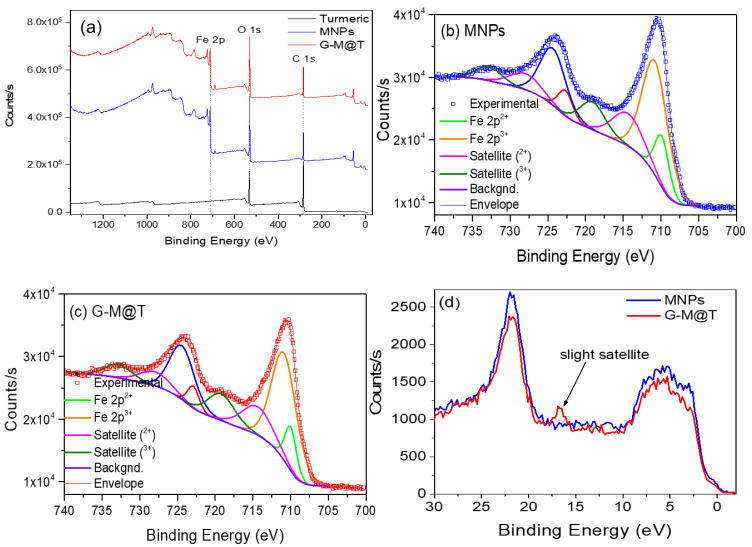
(**a**) XPS survey spectra for turmeric powder, MNPs, and G-M@T samples. XPS core-level spectra of Fe 2p doublets for (**b**) MNPs and (**c**) G-M@T samples. (**d**) XPS valence band spectra of MNPs and G-M@T samples.

**Figure 4 materials-16-03020-f004:**
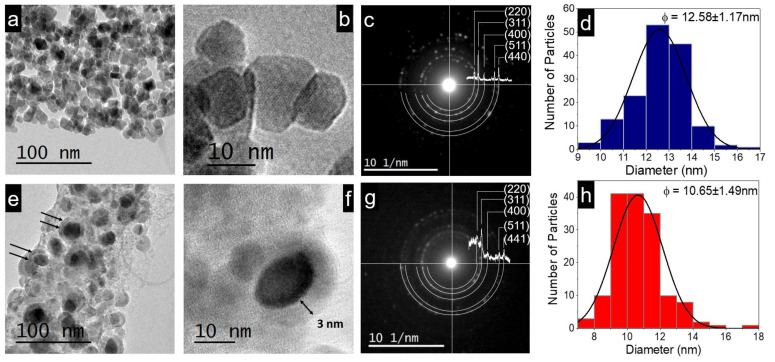
TEM micrographs for (**a**–**d**) MNPs and (**e**–**h**) G-M@T samples. In (**e**), the black arrows indicate the organic cover over the SPIONs. In (**c**,**g**), the SAED results are in agreement with XRD analysis corresponding to magnetite phase as cores (black contrast in the micrograph) and organic shell (gray contrast in (**e**,**f**)). The histograms (**d**,**h**) show the mean particle size fitted to Gaussian functions.

**Figure 5 materials-16-03020-f005:**
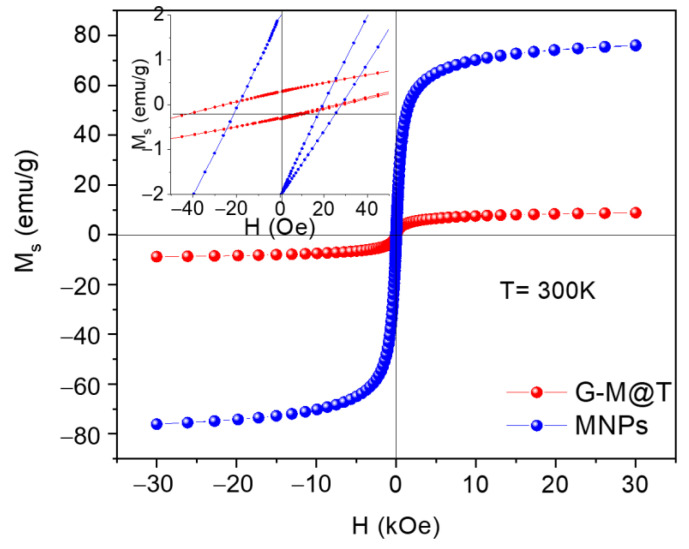
Hysteresis loops of MNPs and G-M@T samples measured at 300 K and a magnetic field of 3 T. The inset shows H_c_ differences due to the nanoparticle surface.

**Figure 6 materials-16-03020-f006:**
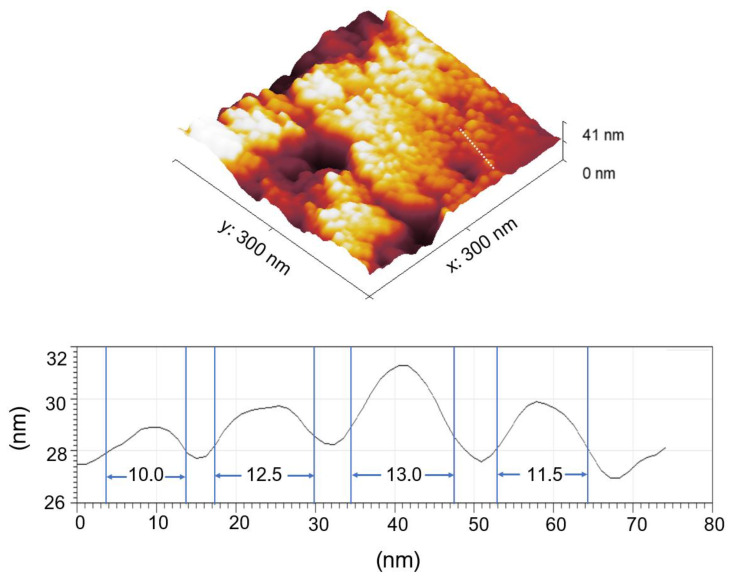
Topography of G-M@T magnetic nanoparticles.

**Figure 7 materials-16-03020-f007:**
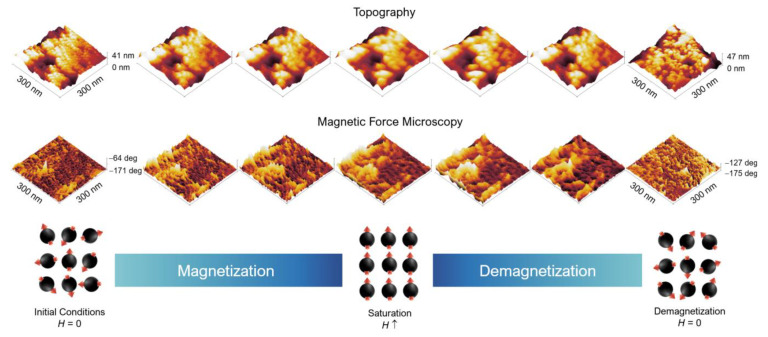
Topography and magnetic domains of G-M@T samples: Magnetic domains in the core region and orientation in response to the AC field, attractive (brighter zones) or repulsive (darker zones) interactions.

**Figure 8 materials-16-03020-f008:**
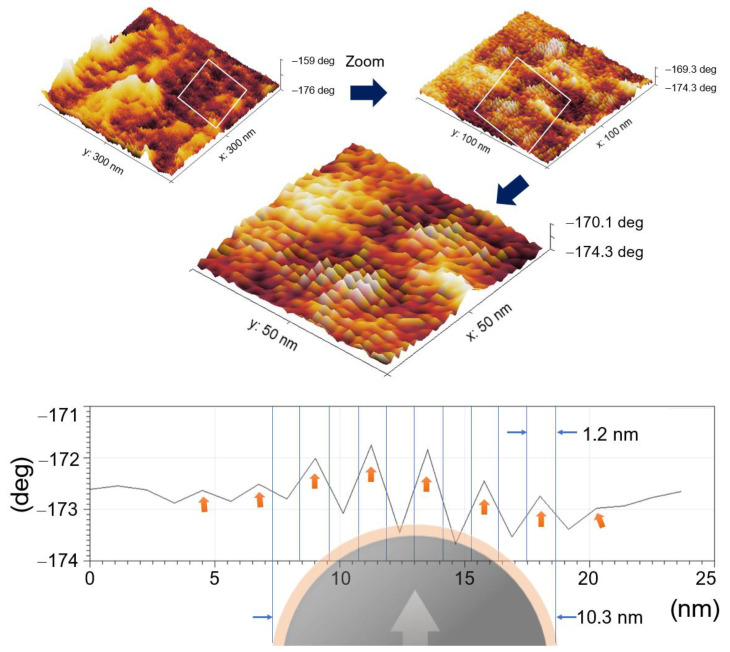
Magnetic domains of G-M@T samples.

**Figure 9 materials-16-03020-f009:**
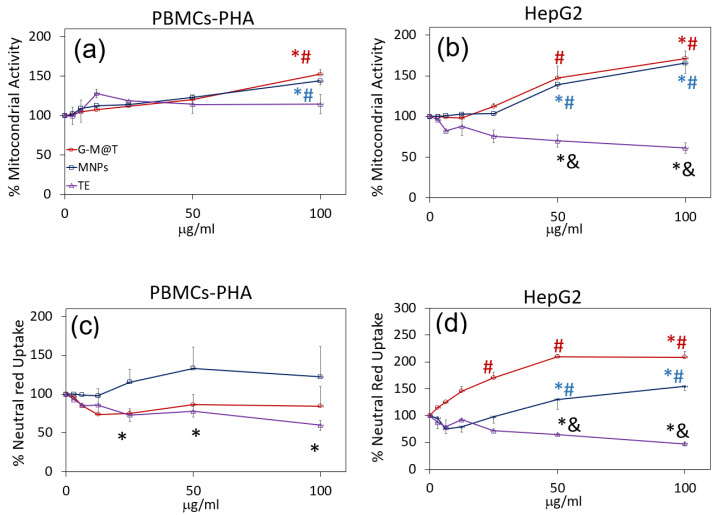
Toxicity of MNPs, G-M@T, and TE in PBMCs (**a**,**c**) and HepG2 cells (**b**,**d**) by MTT assay (**a**,**b**) or Neutra Red Uptake assay (**c**,**d**) after 48 h of exposure. * *p* < 0.05 vs. control, & *p* < 0.05 vs. MNPs, # *p* < 0.05 vs. TE. ANOVA test, *n* = 3 in triplicate.

**Figure 10 materials-16-03020-f010:**
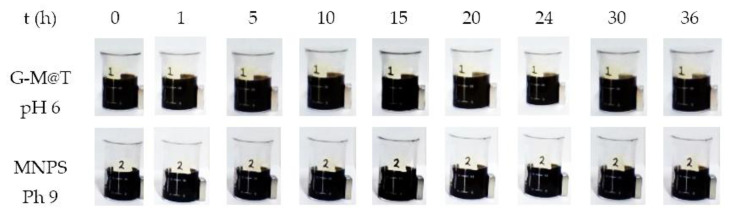
Stability analysis by magnetic precipitation of the nanoparticles synthesized over 36 h.

**Table 1 materials-16-03020-t001:** Structural parameters obtained by XRD.

Sample	Lattice Parameter, *a* (Å)	FWHM ^1^	<*d*> (nm)
MNPs	8.357	0.853	13.57
G-M@T	8.344	1.211	11.70

^1^ FWHM corresponds to the (311) diffraction plane. Theoretical lattice constant *a* = 8.393 Å.

## Data Availability

Data are available upon reasonable request.

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
