# Peer review of "Processing and Physicochemical Properties of Magnetite Nanoparticles Coated with Curcuma longa L. Extract"

_materials, 2023, doi:10.3390/ma16083020_

Round 1
Reviewer 1 Report
This work presents the iron oxide NPs coated with Curcuma Longa rhizome - its synthesis, coating, and characterization. Despite the detailed studies, still, this work needs an improvement to be published. The main goal of application is not clear. Authors propose delivery of TE to affect the cancer cells, but in the manuscript also the band gap studies are mentioned. The Ms values seems to be weak to be applied in magnetic hyperthermia. There is lack of the stability studies, etc.
I would have suggestions and questions that need to be addressed as follows:
1) The extract contains much more compounds than C21H20O6, so using of this formula does not have sense. It would suggest that only a particular compound would be in the extract.
2) Ms value is very low comparing to the bare spions what limits their further use, especially when it comes to the magnetic hyperthermia application.
3) Line 150 - please correct the unit - there is 20 gr. that should be g, other units also need correction e.g. mL should be used, not ml; there is a lack of space between number and unit in many places
4) pH 14 during synthesis does not lead to the Fe3O4 synthesis. Indeed, at the initial range up to pH 11-12 it forms, but then, the hydroxides form.
5) There is a lack of discussion and no results on turmeric extract obtained in this work. So far, no details which compounds were extracted. There might be different compounds, not only this one that is presented on the graphical scheme. There is also no information how the organic compound is bond to the surface of SPIONs.
6) The is a dramatic loss of the Ms values in the samples with TE. The explanation needs to be improved.
7) Authors mention the application of Photoacoustic spectroscopy but show no results in the main text, only cite the work. What is the sense to show the band gap energy in here? What is the final application of this material. If photoinduced processes, the studies need to be extended to study radical mediated processes in the different media.
8) FT-IR analysis is poorly described, especially when it comes to the TE.
9) There is a lack of TGA results. It should be presented what is a mass ratio between the magnetic core and the shell to discuss the magnetic properties, especially low Ms values.
10) No zeta potential studies are revealed. It is not clear if the final product is a stable colloid, if not, there is no sense to deliver it to cells, especially if the tend to aggregate and affect cells.
11) Turbidymetry or other related technique to determine the stability should be also added to this work.
12) Please update the recent articles in MH application and TE delivery.
In my opinion this work should be rejected in this form, but after adding data it can be resubmitted and considered for publication.
Author Response
Dear Reviewer,
We appreciate the referee´s comments and suggestions to improve our manuscript. We try to consider all your observations, being added to the corrected manuscript.
We have attached the answer file .
Thank you.

Reviewer 2 Report
The manuscript “Processing and physicochemical properties of magnetite nanoparticles coated with Curcuma longa L. extract (Fe3O4@C21H20O6)” describes the synthesis of (Fe3O4@C21H20O6) and its biological activity. The manuscript needs minor changes before publication in this journal.
1. In the abstract “Curcuma longa L. is part of the ginger family (Zingiberaceae), and its polyphenol structure has an affinity to be linked to Fe ions” sentence should be changed to “Curcuma longa L. is part of the ginger family (Zingiberaceae), the extracts of this plant contains polyphenol structure compound and it has an affinity to be linked to Fe ions”
2. In line 76- Curcuma longa L. (C96H104O20), remove the formula(C96H104O20). This formula does not match with curcumin structure, and more over the plant contains several compounds in addition to curcumin.
3. In the introduction part, the purpose of this study should be clearly mentioned.
4. This paper describes the importance of Fe3O4@C21H20O6. So, I recommend the author write the synthetic procedure in detail when the KOH solution and turmeric extract should be added.
5. The scientific notation such as “gr” “ml” should be corrected “g” “mL” throughout the manuscript. Also, check the typographical errors.
Author Response
Dear editors y reviewers
We perform the suggested corrections to the manuscript in order to be considered in the Special Issue "Biocompatible Nanostructures: Research, Development, and Applications", Material MDPI.
Best regards,
We thank the referee´s comments and suggestions to improve our manuscript. We try to consider all your observations, being added to the corrected manuscript.
Answers to Reviewer #2
The manuscript “Processing and physicochemical properties of magnetite nanoparticles coated with Curcuma longa L. extract (Fe3O4@C21H20O6)” describes the synthesis of (Fe3O4@C21H20O6) and its biological activity. The manuscript needs minor changes before publication in this journal
- In the abstract “Curcuma longa L. is part of the ginger family (Zingiberaceae), and its polyphenol structure has an affinity to be linked to Fe ions” sentence should be changed to “Curcuma longa L. is part of the ginger family (Zingiberaceae), the extracts of this plant contains polyphenol structure compound and it has an affinity to be linked to Fe ions”
We attended the reviewer´s suggestion and in the corrected manuscript, it was changed that sentence in the abstract section.
- In line 76- Curcuma longa L. (C96H104O20), remove the formula(C96H104O20). This formula does not match with curcumin structure, and more over the plant contains several compounds in addition to curcumin.
By following the referee´s suggestion, we remove the formula (C96H104O20) in 76 line in the corrected manuscript.
- In the introduction part, the purpose of this study should be clearly mentioned.
According to the reviewer´s observation, the following paragraph was added in the corrected manuscript:
In this work the coprecipitation route from the iron salts was improved to obtain MNPs coated directly with TE (Figure 1a), the synthesis of samples was verified, and finally their properties were evaluated as well as their cell viability in HepG2 and PBMCs lines.
- This paper describes the importance of Fe3O4@C21H20O6. So, I recommend the author write the synthetic procedure in detail when the KOH solution and turmeric extract should be added.
Attending the referee´s suggestion, we add the following paragraph in methods section:
As in the co-precipitation route from iron salts for bare MNPs, the synthesis initiates using 6.96 g of FeCl3•6H2O and 2.52 g of FeCl2•4H2O mixed separately during 30 min with distilled water in a molarity of 0.02 M and 0.01M respectively. N2 were injected in a 3-necked flask during all reaction procedure and heated at 70° C in a water bath. Then, ½ Fe+2: Fe+3 solution was poured into the 3-necked flask and stirred mechanically at 200 rpm for 15 min. 10 mL of 7 M KOH in deionized water was added to the 3-necked flask by dropping.
After 10 min, 50 mL of TE was dropped in the solution, reaching pH = 14, and a black colour solution. The synthesis was poured into a beaker allowing it to cool, removing the supernatant. 50 mL of ethanol and 50 mL of deionized water were added, sonicating the mixture for 5 min, and letting the solution container rest on a magnet until the sample precipitated. Two washes using ethanol/water solution were per-formed and four more washes were carried out with double distilled water, lyophilizing the sample at the end, and being stored under vacuum conditions. The G-M@T powder obtained was also sensitive to magnetic field.
- The scientific notation such as “gr” “ml” should be corrected “g” “mL” throughout the manuscript. Also, check the typographical errors.
We attended the reviewer´s suggestion and these errors were corrected of the manuscript
Reviewer 3 Report
The authors have synthesized the Fe3O4@C21H20O6 and investigated the material characterization of these composite samples. As a whole, the topic is interesting, the data is well characterized, and is explained well. I recommended this work for publication in Materials. I have the following comments, which may be considered before accepting the manuscript for publication.
1. Some references about the previous topics of ferrite composites with various applications may be added in the revision to extend the readership, e.g. DOI: https://doi.org/10.1016/j.ceramint.2022.03.307; https://doi.org/10.1016/j.matchemphys.2015.12.065 ; https://doi.org/10.1016/j.micromeso.2019.109823
2. The author stated in the title that the content of Curcuma longa L. extract was only curcumin (C21H20O6). In the introduction section, the author explains that curcumin is only about 77% of the total content and contains other compounds. Besides, the author also mentioned the compound formula C96H104O20 in the introduction for the Turmeric sample. How can the author decide to use the compound formula Fe3O4@C21H20O6, not Fe3O4@C96H104O20?
3. The author mentions that Commercial turmeric powder from India (TRS, Curcuma longa L. Powder) was used in this study. However, in the method section (section 2.2), Turmeric extract preparation is carried out. Is this extraction done further to get only the curcumin compound (C21H20O6) from this powder? I suggest adding the title of this subchapter for the extraction of what compounds.
4. It is possible for the authors to add the reaction mechanism that occurs from the precursor to the oxide product with the addition of each substance.
Author Response
Dear referees:
We thank the referee´s comments and suggestions to improve our manuscript. We try to consider all your observations, being added to the corrected manuscript.
Answers to Reviewer #3
The authors have synthesized the Fe3O4@C21H20O6 and investigated the material characterization of these composite samples. As a whole, the topic is interesting, the data is well characterized, and is explained well. I recommended this work for publication in Materials. I have the following comments, which may be considered before accepting the manuscript for publication.
- Some references about the previous topics of ferrite composites with various applications may be added in the revision to extend the readership, e.g. DOI: https://doi.org/10.1016/j.ceramint.2022.03.307;https://doi.org/10.1016/j.matchemphys.2015.12.065; https://doi.org/10.1016/j.micromeso.2019.109823
By following the referee´s suggestion, we add these and more references about various Fe3O4 nanoparticles applications, adding the following paragraph in the introduction section.
Currently, iron oxide superparamagnetic nanoparticles (SPIONs) have been studied for their scientific and technological applications, such as electronic devices, the food industry, the environment, and medical fields [1–8].
- The author stated in the title that the content of Curcuma longa L. extract was only curcumin (C21H20O6). In the introduction section, the author explains that curcumin is only about 77% of the total content and contains other compounds. Besides, the author also mentioned the compound formula C96H104O20 in the introduction for the Turmeric sample. How can the author decide to use the compound formula Fe3O4@C21H20O6, not Fe3O4@C96H104O20
Thank you for your observation, indeed Curcuma longa L. contains principally the following polyphenols:
Curcumin (as major component 77%) demethoxycurcumin (DMC) (17%) and bisdemethoxycurcumin (BDMC) (3%) [9–12]. In the turmeric extraction, these polyphenols are also presents [13], then, the formula C21H20O6 was eliminated from the manuscript, even from the title, to avoid confusion. Also, the following paragraphs were added in the introduction and methods section of the manuscript:
In addition to curcumin, demethoxycurcumin and bisdemethoxycurcumin have been recognized as anti-tumor, anti-cancer, anti-inflammatory and as effective antioxidants, reporting antioxidant activities of 81.98%, 81.77%, and 73% respectively [14–18].
For this extraction, the method described by Alvis et al., 2012 was used with the purpose of preserving the highest concentration of phenolic (curcumin, desmethoxycurcumin and bisdemethoxycurcumin) and antioxidant compounds, maintaining its free radical scavenger activity in the resulting extract [13].
Also figure 1 and graphical abstract where modified attending to the reviewer suggestion.
- The author mentions that Commercial turmeric powder from India (TRS, Curcuma longa L. Powder) was used in this study. However, in the method section (section 2.2), Turmeric extract preparation is carried out. Is this extraction done further to get only the curcumin compound (C21H20O6) from this powder? I suggest adding the title of this subchapter for the extraction of what compounds.
By following the referee´s suggestion, we add the information requested in methods section:
Turmeric extract preparation
For this extraction, the method described by Alvis et al., 2012 was used with the purpose of preserving the highest concentration of phenolic (curcumin, desmethoxycurcumin and bisdemethoxycurcumin) and antioxidant compounds, maintaining its free radical scavenger activity in the resulting extract [13].
First, 20 g. of commercial turmeric powder was mixed in 400 mL of ethanol 79%, heated at 200 °C, stirred magnetically at 400 rpm for 20 min and allowed to cool [13].
- It is possible for the authors to add the reaction mechanism that occurs from the precursor to the oxide product with the addition of each substance.
Attending the referee´s suggestion, we add the following description in Introduction section:
The phytochemicals add organic chains which play a major role in reducing precursor salts to nanomaterials[19]. The mechanism to induce the direct formation of Fe3O4 covered is described as follows: It was use FeCl2•4H2O and FeCl3•6H2O as precursors in the synthesis, then the base KOH was added, which acts as a reductor and co-precipitation agent. At basic pH (>9), -OH groups oxidize, donating their electrons and forming a complex with metallic cations, reducing Fe2+/Fe3+ ions [20,21]. Finally, TE is added to the reaction reaching a saturated pH (10-14) and a black color solution.
It has been reported that, from pH = 10 up to 14, Fe3O4 nanoparticles can be synthesized by co-precipitation route with Fe3+ to Fe2+ ratio of 2:1 in the presence of non-oxidizing conditions [1,22–24]. Also, a systematic study reports that when the pH increases, the size decreases [1].
The delocalized electrons present in the aromatic ring of TE are donated too, potentiating the reduction reaction, and due to the hydroxyl -OH or carboxyl -COOH bonds present in their polyphenols, the direct organic surface encapsulation of magnetite is favored by a covalent iron binding, via ortho-dihydroxy (catechol) or trihydroxy benzene [20,25].
During the reduction, the organic acids carboxyl groups (-COOH) lose their hydrogen atom and become carboxylate ions (COO−). This mechanism stabilizes the electrosterical attachment to the surface of the nanoparticle through organic chains [3,20].
Dear editors y reviewers
We perform the suggested corrections to the manuscript in order to be considered in the Special Issue "Biocompatible Nanostructures: Research, Development, and Applications", Material MDPI.
Best regards,
- Santoyo Salazar, J.; Perez, L.; de Abril, O.; Truong Phuoc, L.; Ihiawakrim, D.; Vazquez, M.; Greneche, J.-M.; Begin-Colin, S.; Pourroy, G. Magnetic Iron Oxide Nanoparticles in 10−40 Nm Range: Composition in Terms of Magnetite/Maghemite Ratio and Effect on the Magnetic Properties. Chemistry of Materials 2011, 23, 1379–1386, doi:10.1021/cm103188a.
- Yew, Y.P.; Shameli, K.; Miyake, M.; Ahmad Khairudin, N.B.B.; Mohamad, S.E.B.; Naiki, T.; Lee, K.X. Green Biosynthesis of Superparamagnetic Magnetite Fe3O4 Nanoparticles and Biomedical Applications in Targeted Anticancer Drug Delivery System: A Review. Arabian Journal of Chemistry 2020, 13, 2287–2308.
- Liandi, A.R.; Cahyana, A.H.; Yunarti, R.T.; Wendari, T.P. Facile Synthesis of Magnetic Fe3O4@Chitosan Nanocomposite as Environmentally Green Catalyst in Multicomponent Knoevenagel-Michael Domino Reaction. Ceram Int 2022, 48, 20266–20274, doi:10.1016/j.ceramint.2022.03.307.
- Thu Huong, L.T.; Nam, N.H.; Doan, D.H.; My Nhung, H.T.; Quang, B.T.; Nam, P.H.; Thong, P.Q.; Phuc, N.X.; Thu, H.P. Folate Attached, Curcumin Loaded Fe3O4 Nanoparticles: A Novel Multifunctional Drug Delivery System for Cancer Treatment. Mater Chem Phys 2016, 172, 98–104, doi:10.1016/j.matchemphys.2015.12.065.
- Tao, C.; Chen, T.; Liu, H.; Su, S. Design of Biocompatible Fe3O4@MPDA Mesoporous Core-Shell Nanospheres for Drug Delivery. Microporous and Mesoporous Materials 2020, 293, doi:10.1016/j.micromeso.2019.109823.
- Pazouki, N.; Irani, S.; Olov, N.; Atyabi, S.M.; Bagheri-Khoulenjani, S. Fe3O4 Nanoparticles Coated with Carboxymethyl Chitosan Containing Curcumin in Combination with Hyperthermia Induced Apoptosis in Breast Cancer Cells. Prog Biomater 2022, 11, 43–54, doi:10.1007/s40204-021-00178-z.
- Wu, K.; Mohsin, A.; Zaman, W.Q.; Zhang, Z.; Guan, W.; Chu, M.; Zhuang, Y.; Guo, M. Urchin-like Magnetic Microspheres for Cancer Therapy through Synergistic Effect of Mechanical Force, Photothermal and Photodynamic Effects. J Nanobiotechnology 2022, 20, doi:10.1186/s12951-022-01411-y.
- Schneider, M.G.M.; Martín, M.J.; Otarola, J.; Vakarelska, E.; Simeonov, V.; Lassalle, V.; Nedyalkova, M. Biomedical Applications of Iron Oxide Nanoparticles: Current Insights Progress and Perspectives. Pharmaceutics 2022, 14.
- Akaberi, M.; Sahebkar, A.; Emami, S.A. Turmeric and Curcumin: From Traditional to Modern Medicine. In Advances in Experimental Medicine and Biology; Springer, 2021; Vol. 1291, pp. 15–39.
- Vafaeipour, Z.; Razavi, B.M.; Hosseinzadeh, H. Effects of Turmeric (Curcuma Longa) and Its Constituent (Curcumin) on the Metabolic Syndrome: An Updated Review. J Integr Med 2022, 20, 193–203, doi:10.1016/j.joim.2022.02.008.
- Fatima, G.; Loubna, A.; Wiame, L.; Azeddine, I. In Silico Inhibition Studies of AXL Kinase by Curcumin and Its Natural Derivatives. J Appl Bioinforma Comput Biol 2017, 06, doi:10.4172/2329-9533.1000142.
- Beyene, A.M.; Moniruzzaman, M.; Karthikeyan, A.; Min, T. Curcumin Nanoformulations with Metal Oxide Nanomaterials for Biomedical Applications. Nanomaterials 2021, 11, 460, doi:10.3390/nano11020460.
- Alvis, A.; Arrazola, G.; Martinez, W. Evaluación de La Actividad y El Potencial Antioxidante de Extractos Hidro-Alcohólicos de Cúrcuma (Cúrcuma Longa). Información tecnológica 2012, 23, 11–18, doi:10.4067/S0718-07642012000200003.
- Jayaprakasha, G.K.; Jaganmohan Rao, L.; Sakariah, K.K. Antioxidant Activities of Curcumin, Demethoxycurcumin and Bisdemethoxycurcumin. Food Chem 2006, 98, 720–724, doi:10.1016/j.foodchem.2005.06.037.
- Hatamipour, M.; Ramezani, M.; Tabassi, S.A.S.; Johnston, T.P.; Sahebkar, A. Demethoxycurcumin: A Naturally Occurring Curcumin Analogue for Treating Non-Cancerous Diseases. J Cell Physiol 2019, 234, 19320–19330.
- Lee, Y.S.; Oh, S.M.; Li, Q.Q.; Kim, K.W.; Yoon, D.; Lee, M.H.; Kwon, D.Y.; Kang, O.H.; Lee, D.Y. Validation of a Quantification Method for Curcumin Derivatives and Their Hepatoprotective Effects on Nonalcoholic Fatty Liver Disease. Curr Issues Mol Biol 2022, 44, 409–432, doi:10.3390/cimb44010029.
- Xiang, M.; Jiang, H.G.; Shu, Y.; Chen, Y.J.; Jin, J.; Zhu, Y.M.; Li, M.Y.; Wu, J.N.; Li, J. Bisdemethoxycurcumin Enhances the Sensitivity of Non-Small Cell Lung Cancer Cells to Icotinib via Dual Induction of Autophagy and Apoptosis. Int J Biol Sci 2020, 16, 1536–1550, doi:10.7150/ijbs.40042.
- Kim, S.B.; Kang, O.H.; Lee, Y.S.; Han, S.H.; Ahn, Y.S.; Cha, S.W.; Seo, Y.S.; Kong, R.; Kwon, D.Y. Hepatoprotective Effect and Synergism of Bisdemethoycurcumin against MCD Diet-Induced Nonalcoholic Fatty Liver Disease in Mice. PLoS One 2016, 11, doi:10.1371/journal.pone.0147745.
- Sundaramurthy, A. Phytosynthesized Nanomaterials—NextGen Material for Biomedical Applications. In Emerging Phytosynthesized Nanomaterials for Biomedical Applications; Elsevier, 2023; pp. 31–64.
- Ramirez-Nuñez, A.L.; Jimenez-Garcia, L.F.; Goya, G.F.; Sanz, B.; Santoyo-Salazar, J. In Vitro Magnetic Hyperthermia Using Polyphenol-Coated Fe 3 O 4 @ γ Fe 2 O 3 Nanoparticles from Cinnamomun Verum and Vanilla Planifolia : The Concert of Green Synthesis and Therapeutic Possibilities. Nanotechnology 2018, 29, 074001, doi:10.1088/1361-6528/aaa2c1.
- Massart, R. Preparation of Aqueous Magnetic Liquids in Alkaline and Acidic Media. IEEE Trans Magn 1981, 17, 1247–1248, doi:10.1109/TMAG.1981.1061188.
- Singh, P.; Sharma, K.; Hasija, V.; Sharma, V.; Sharma, S.; Raizada, P.; Singh, M.; Saini, A.K.; Hosseini-Bandegharaei, A.; Thakur, V.K. Systematic Review on Applicability of Magnetic Iron Oxides–Integrated Photocatalysts for Degradation of Organic Pollutants in Water. Mater Today Chem 2019, 14.
- Salehipour, M.; Rezaei, S.; Mosafer, J.; Pakdin-Parizi, Z.; Motaharian, A.; Mogharabi-Manzari, M. Recent Advances in Polymer-Coated Iron Oxide Nanoparticles as Magnetic Resonance Imaging Contrast Agents. Journal of Nanoparticle Research 2021, 23.
- Aisida, S.O.; Akpa, P.A.; Ahmad, I.; Zhao, T. kai; Maaza, M.; Ezema, F.I. Bio-Inspired Encapsulation and Functionalization of Iron Oxide Nanoparticles for Biomedical Applications. Eur Polym J 2020, 122.
- Herrera-Becerra, R.; Zorrilla, C.; Ascencio, J.A. Production of Iron Oxide Nanoparticles by a Biosynthesis Method: An Environmentally Friendly Route. Journal of Physical Chemistry C 2007, 111, 16147–16153, doi:10.1021/jp072259a.
Round 2
Reviewer 1 Report
Authors did not address the comments relating to the required analyses. For example, some results like estimation of the extract content are only assumed by calculations without delivering experimental data. The other parts of the papers still needs improvement. In my opinion, the paper should be rejected.
Author Response
Dear Reviewer,
We appreciate the referee´s comments and suggestions to improve our manuscript. We consider all your observations to be added to the corrected manuscript.
We have attached the answer file.
Thank you.

Reviewer 3 Report
I am satisfied that the authors have revised their manuscript in line with all reviewer comments and it is now suitable for publication in Materials.
Author Response
Dear Reviewer,
We appreciate the referee´s comments and suggestions to improve our manuscript.
Thank you in advance.
